# Comprehensive Subset Selection for CT Volume Compression to Improve Pulmonary Disease Screening Efficiency

Qian Shao*
12221112@zju.edu.cn
Zhejiang University
Hangzhou, Zhejiang, China

Kai Zhang*
zhangkai1999@zju.edu.cn
Zhejiang University
Hangzhou, Zhejiang, China

Bang Du
bangdu1994@zju.edu.cn
Zhejiang University
Hangzhou, Zhejiang, China

Zepeng Li
lizepeng@zju.edu.cn
Zhejiang University
Hangzhou, Zhejiang, China

Yixuan Wu
wyx_chloe@zju.edu.cn
Zhejiang University
Hangzhou, Zhejiang, China

Qiyuan Chen
chenqiyuan1012@foxmail.com
Zhejiang University
Hangzhou, Zhejiang, China

Jian Wu†
wujian2000@zju.edu.cn
School of Public Health Zhejiang
University
Hangzhou, Zhejiang, China

Jintai Chen†
jtchen721@gmail.com
University of Illinois at
Urbana-Champaign
Champaign, Illinois, United States

## ABSTRACT

Deep learning models are widely used to process Computed Tomography (CT) data in the automated screening of pulmonary diseases, significantly reducing the workload of physicians. However, the three-dimensional nature of CT volumes involves an excessive number of voxels, which significantly increases the complexity of model processing. Previous screening approaches often overlook this issue, which undoubtedly reduces screening efficiency. Towards efficient and effective screening, we design a hierarchical approach to reduce the computational cost of pulmonary disease screening. The new approach re-organizes the screening workflows into three steps. First, we propose a Computed Tomography Volume Compression (CTVC) method to select a small slice subset that comprehensively represents the whole CT volume. Second, the selected CT slices are used to detect pulmonary diseases coarsely via a lightweight classification model. Third, an uncertainty measurement strategy is applied to identify samples with low diagnostic confidence, which are re-detected by radiologists. Experiments on two public pulmonary disease datasets demonstrate that our approach achieves comparable accuracy and recall while requiring approximately 4.5% of the time needed by the counterparts using full CT volumes. Besides, we also found that our approach outperforms previous cutting-edge CTVC methods in retaining important indications after compression.

*Both authors contributed equally to this research.
†Corresponding Author.

## CCS CONCEPTS

• **Computing methodologies → Computer vision**.

## KEYWORDS

Pulmonary disease screening, Computed tomography, Comprehensive subset selection

**ACM Reference Format:**
Qian Shao, Kai Zhang, Bang Du, Zepeng Li, Yixuan Wu, Qiyuan Chen, Jian Wu, and Jintai Chen. 2024. Comprehensive Subset Selection for CT Volume Compression to Improve Pulmonary Disease Screening Efficiency. In *Proceedings of Artificial Intelligence and Data Science for Healthcare: Bridging Data-Centric AI and People-Centric Healthcare (AIDSH 2024)*. ACM, New York, NY, USA, 7 pages. https://doi.org/XXXXXXX.XXXXXXX

## 1 INTRODUCTION

With the rapid advancements in medical imaging and deep learning algorithms, more researchers are utilizing deep learning methods to achieve expert-level disease screening, significantly reducing physicians' workloads [3]. Computed Tomography (CT) data are extensively used for diagnosing and screening various pulmonary diseases, such as lung carcinoma and pneumonia, due to CT's ability to reveal crucial details specific to individual patients [11]. However, pulmonary CT volumes, consisting of dozens or even hundreds of individual slices, present high computational complexity and require substantial memory capacity for processing—resources that are often limited in real-world medical settings.

To efficiently and accurately screen for pulmonary diseases, we design a novel screening approach. The core methodology insight is leveraging a Comprehensive Subset Selection (CSS) method to select representative slices for coarse detection, and then, during the inference phase, we devise an uncertainty strategy to identify samples with low diagnostic confidence. Radiologists will re-detect these samples. Our proposed CSS method ensures that the selected slices represent the entire CT volume, reducing the likelihood of omitting slices containing crucial indicators. While traditional slice

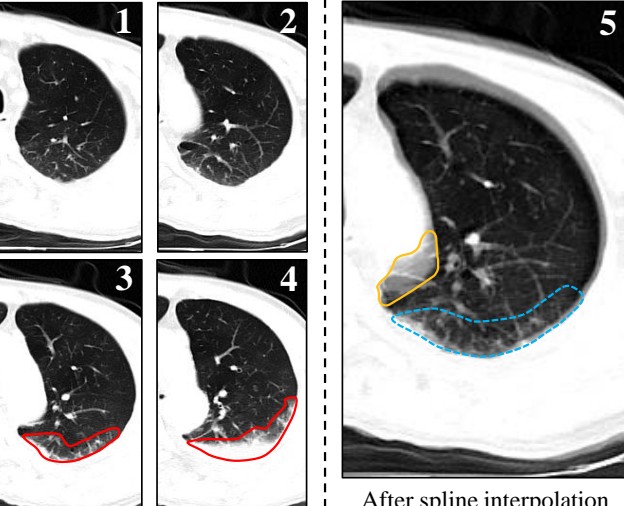

After spline interpolation

**Figure 1: Visualization results of interpolation-based methods. Four pictures on the left represent four consecutive slices that contain CP lesions. The right picture illustrates the visualization results obtained from the spline interpolation. Within these visualizations, areas enclosed by red solid lines represent the CP lesions, while regions within yellow solid lines indicate artefacts generated after processing. And areas enclosed by blue dashed lines depict regions where lesions were eliminated after processing.**

selection methods [26, 27] cannot always guarantee this. In addition, our method also avoids the problem of introducing artefacts that interpolation-based methods usually face [18, 5, 13], as shown in Fig.1.

We conduct comparative experiments on two publicly available CT datasets to verify the accuracy and efficiency of our proposed screening approach, containing screening tasks for three types of pulmonary diseases, with a total of 2654 CT volumes. Specifically, we train classification networks using CT data compressed by different methods and compare their accuracy and recall. The experimental results demonstrate that our proposed CSS method significantly outperforms other methods. Furthermore, we explore the performance of the classification model under extreme conditions. For instance, when retaining only 4 slices from a single CT volume, the classification model still achieves a recall of 95.19% in the screening task for COVID-19. Based on this, we apply the uncertainty strategy to select low-confidence samples for re-detection, further improving the recall and meeting the clinical diagnosis standards. This proves that the uncertainty strategy screens out low-confidence cases that contain most of the misdiagnosed samples, which can provide reference value for actual disease screening tasks. Finally, we compare the inference time of different methods, validating the efficiency of our approach.

In summary, the main contributions of this article are as follows:

(1) We design a novel hierarchical approach based on CT data for accurate and efficient screening of pulmonary diseases.

(2) We propose a slice selection method CSS, which can select a representative slice subset from the whole CT volume, thereby enabling efficient screening.

(3) We adopt an uncertainty strategy to select low-confidence samples for re-detection during the inference phase, ensuring the accuracy of the screening.

(4) We validate the efficiency and accuracy of our proposed screening system on two public CT datasets. Experiments show that our screening method can achieve similar performance while requiring approximately 4.5% of the time needed by the method using full CT volumes.

## 2 RELATED WORK

Numerous studies have proposed various methods to process CT volumes for efficient screening of pulmonary diseases, which can be broadly categorized into three types:

*Patch-based methods.* These methods divide CT volumes into smaller patches for processing [8], which is similar to serializing parallel tasks. While this approach reduces the demand for large memory, it prolongs the inference time, resulting in inefficiency.

*Slice selection-based methods.* These methods select a subset of slices from the CT volume evenly to construct the desired volume [26, 27]. However, this approach lacks assurance that the selected slices represent the entire CT volume, as there is a high likelihood of omitting slices containing crucial indicators of the disease, leading to inaccurate screening results.

*Interpolation-based methods.* These methods adjust the Z-axis by either compressing or expanding it to reach the desired depth, such as Linear Interpolation (LI) [16, 18], Spline Interpolation (SI) [2, 5, 6], or Projection Interpolation (PI) [13]. However, these methods may distort the original pixel values, potentially introducing artefacts or losing lesion information, inevitably affecting screening accuracy.

## 3 METHODOLOGY

The overall process of our screening approach is divided into three steps, as shown in Fig. 2: First, we design CSS to select a slice subset that represents the entire CT volume. Second, we train a classification network to detect disease using the selected slices. Third, we adopt the uncertainty strategy to select low-confidence samples for re-detection.

### 3.1 Comprehensive Subset Selection

Assuming that each CT volume contains $n$ slices (denoted as $\{\mathbf{x}_i\}_{i=1}^{n}$), our goal is to select $m(\ll n)$ slices that could represent the whole CT volume so that the performance of a model trained on the selected slice subsets approaches that on the whole CT volumes. The procedure of CSS is shown in Fig. 3.

*3.1.1 Representation Learning.* We leverage the pre-trained image feature extraction capabilities of MedCLIP-ViT [21] to obtain semantically meaningful representations containing key indicators. The image encoder of MedCLIP-ViT maps $\mathbf{x}_i$ onto a $d$-dimensional hypersphere with $L^2$ normalization, denoted as $\mathbf{f}_i = f(\mathbf{x}_i)$. Specifically, the [CLS] token features produced by the model's final output

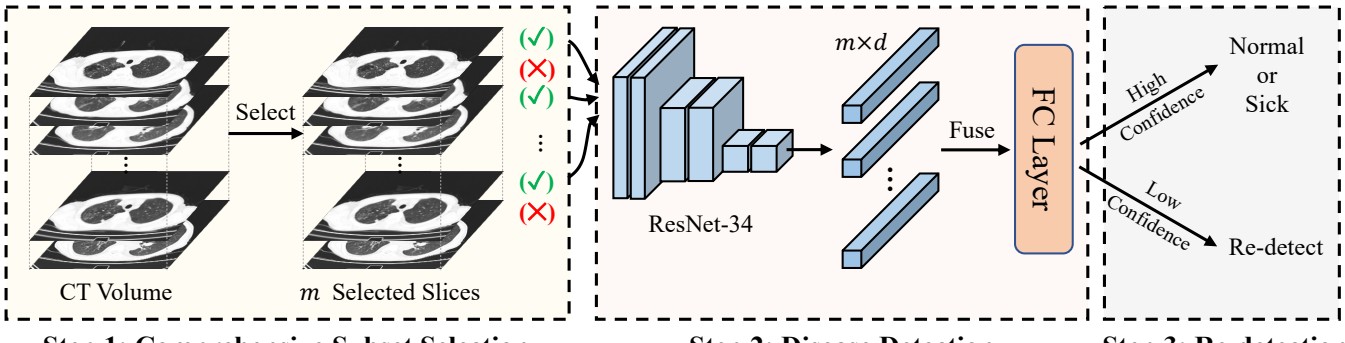

**Figure 2: The overall process of our proposed screening approach.**

with $L^2$ normalization are employed for slice selection. It is noteworthy that although we utilize the larger model MedCLIP-ViT for feature extraction, which incurs additional computational resource consumption, this process is conducted before training the classification models. The efficiency should be evaluated based on the inference time of the models.

*3.1.2 Slice Selection.* We select CT slices according to two strategies that are representativeness and diversity [23, 22, 20, 25]. The former ensures that the selected slice subset can represent the whole CT volume containing key indicators. The latter minimizes the internal similarity of the selected subset to avoid redundancy, which provides the possibility of extremely compressing the CT volume.

First, we adopt $K$-Means clustering that partitions $n$ instances (*i.e.*, the CT slices in the feature space, denoted as $\{\mathbf{f}_i\}_{i=1}^n$) into $m(\leq n)$ clusters, with each cluster represented by its centroid $\mathbf{c}$ and every instance assigned to the cluster of the nearest centroid [4, 14]. Formally, we aim to find $m$-cluster partitioning $\mathbb{S} = \{S_1, S_2, ..., S_m\}$ that minimizes the within-cluster sum of squares [9]:

$$\min_{\mathbb{S}} \sum_{i=1}^m \sum_{\mathbf{f} \in S_i} \|\mathbf{f} - \mathbf{c}_i\|^2 = \min_{\mathbb{S}} \sum_{i=1}^m |S_i| \operatorname{Var}(S_i). \quad (1)$$

It is optimized iteratively with EM [15] from random initial centroids.

Then, we select the density peaks within each cluster as the most representative slices since the density peaks are similar to as many instances as possible. The density is estimated by $K$-Nearest Neighbor ($K$-NN) [17] which is formulated as

$$\operatorname{Den}(\mathbf{f}_i, k) = \frac{k}{n} \frac{1}{A_d \cdot D(\mathbf{f}_i, \mathbf{f}_i^k)}, \quad (2)$$

where $A_d = \pi^{d/2} / \Gamma(\frac{d}{2} + 1)$ is the volume of a unit $d$-dimensional ball, $d$ is the feature dimension, $\Gamma(x)$ is the Gamma function, and $D(\mathbf{f}_i, \mathbf{f}_i^k) = \|\mathbf{f}_i - \mathbf{f}_i^k\|$ is the feature distance between two instances, $\mathbf{f}_i^k$ is the $k$th nearest neighbor of $\mathbf{f}_i$. Intuitively, the smaller the feature distance, the greater the similarity of the instances. However, $\operatorname{Den}(\cdot, \cdot)$ is very sensitive to noise, as it only takes the $k$th nearest neighbour into account. For robustness, we replace the $k$th neighbor distance $D(\mathbf{f}_i, \mathbf{f}_i^k)$ with the average distance $\overline{D}(\mathbf{f}_i, k)$ to all $k$ nearest

neighbors instead:

$$\hat{\operatorname{Den}}(\mathbf{f}_i, k) = \frac{k}{n} \frac{1}{A_d \cdot \overline{D}(\mathbf{f}_i, k)}, \quad (3)$$

where $\overline{D}(\mathbf{f}_i, k) = \frac{1}{k} \sum_{j=1}^k D(\mathbf{f}_i, \mathbf{f}_i^j)$. To sum up, $\hat{\operatorname{Den}}(\mathbf{f}_i, k)$ is used to measure the representativeness of instance $\mathbf{f}_i$. Since only the relative ordering matters in our selection process, the density peak corresponds to the instance with maximum $\hat{\operatorname{Den}}(\mathbf{f}_i, k)$, *i.e.*, minimum $\overline{D}(\mathbf{f}_i, k)$.

However, when the clustering boundaries are located in high-density areas, the selected instances may align along these boundaries and become proximal to each other, leading to redundancy. Therefore, we apply a regularizer to diversify the selected instances in the feature space iteratively. In detail, let $\hat{\mathbb{F}}^t = \{\hat{\mathbf{f}}_1^t, \hat{\mathbf{f}}_2^t, ..., \hat{\mathbf{f}}_m^t\}$ denote the set of $m$ instances selected at iteration $t$, $\hat{\mathbf{f}}_i^t$ is selected from clusters $S_i$, where $i \in \{1, 2, ..., m\}$. For each candidate $\mathbf{f}_i$ in cluster $S_i$, the farther it is away from those in other clusters in $\hat{\mathbb{F}}^{t-1}$, the more diversity it creates. We thus minimize the total inverse distance to others in a regularization loss $\operatorname{Reg}(\mathbf{f}_i, t)$, with a sensitivity hyperparameter $\alpha$:

$$\operatorname{Reg}(\mathbf{f}_i, t) = \sum_{j \neq i} \frac{1}{\|\mathbf{f}_i - \hat{\mathbf{f}}_j^{t-1}\|^\alpha}. \quad (4)$$

This regularizer is updated with an exponential moving average [10]:

$$\overline{\operatorname{Reg}}(\mathbf{f}_i, t) = m_{\operatorname{reg}} \cdot \overline{\operatorname{Reg}}(\mathbf{f}_i, t - 1) + (1 - m_{\operatorname{reg}}) \cdot \operatorname{Reg}(\mathbf{f}_i, t), \quad (5)$$

where $m_{\operatorname{reg}}$ is the momentum. At iteration $t$, we select instance $i$ of the maximum combination of representativeness and diversity $C(\mathbf{f}_i, t)$ within each cluster:

$$C(\mathbf{f}_i, t) = \frac{1}{\overline{D}(\mathbf{f}_i, k)} - \lambda \cdot \overline{\operatorname{Reg}}(\mathbf{f}_i, t), \quad (6)$$

where $\lambda$ is a hyperparameter that balances diversity and representativeness. At the last iteration, the instances in $\hat{\mathbb{F}}$ are the selected CT slices for the downstream task.

## 3.2 Disease Detection

We train a classification model using the selected slices for disease detection. First, ResNet-34 [7] is used to extract image features of $m$ slices in each case, denoted as $f_{Res}(\mathbf{x}_i)$, where $i \in \{1, 2, ..., m\}$. Then, a light transformer-based network [19] is adopted to fuse

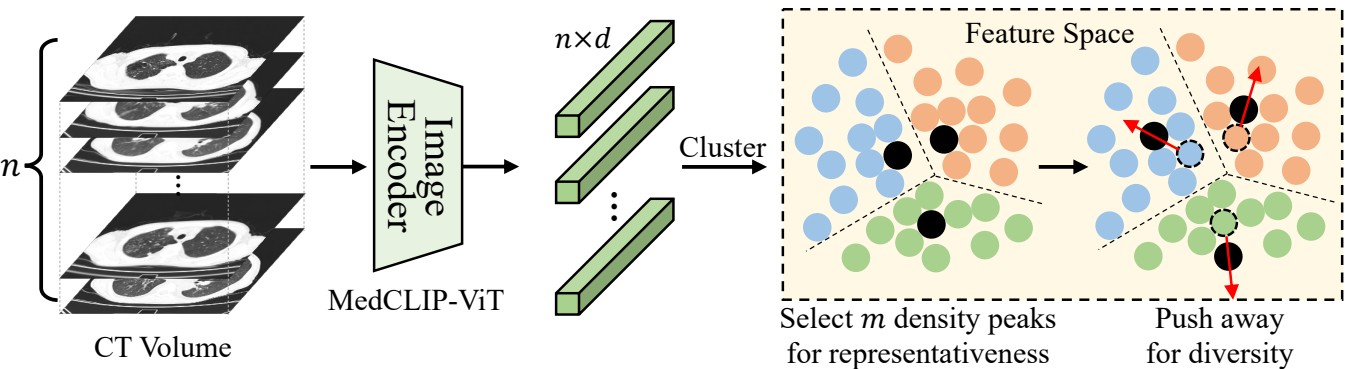

**Figure 3: The procedure of CSS. The dark circles represent the selected slices.**

these features. Finally, we perform a fully connected layer and Softmax on the output of the transformer-based network to predict the probability of disease. Details of the model can be found in Appendix A.

### 3.3 Low-confidence Case Re-detection

Low-confidence cases are re-detected using their corresponding full CT volumes by radiologists, specifically, those where the model's predicted probability of disease is between 0.4 and 0.6. Although this step reduces some screening efficiency, it significantly improves accuracy to meet clinical requirements.

## 4 EXPERIMENTS

### 4.1 Dataset

We conduct comparative experiments on two publicly available datasets: SARS-CoV-2 dataset [24] and LUNG-PET-CT-Dx [12], which are obtained from the China Consortium of Chest CT Image Investigation (CC-CCII) and The Cancer Imaging Archive (TCIA) [1], respectively. Given our objective to compress CT volumes, we eliminated cases with a relatively low number of slices, specifically those with less than 64 slices. After that, 2654 cases are included in the study, containing 311 cases diagnosed with adenocarcinoma (AC), 747 cases with Novel Coronavirus Pneumonia (NCP, *i.e.,* COVID-19), 741 cases with Common Pneumonia (CP) and 855 normal cases. Here, a case refers to a 3D CT volume. The AC cases are from LUNG-PET-CT-Dx, while the NCP, CP and normal cases are from the SARS-CoV-2 dataset. Then, we design three binary classification tasks to detect the above three diseases. Specifically, Task 1 contains 747 NCP and 855 normal cases; Task 2 contains 741 CP and 855 normal cases; Task 3 contains 311 AC and 311 normal cases. The normal cases for Task 3 are randomly chosen from 855 normal cases of the SARS-CoV-2 dataset. Finally, we evenly divide the dataset into 5 parts and use 5-fold cross-validation to evaluate the accuracy and recall for each task.

### 4.2 Experimental Settings

All models are implemented in PyTorch and trained on an RTX 4090 with 24 GB memory. Considering the anatomical structure of the lungs—where the left lung is divided into upper and lower lobes with a volume ratio of approximately 0.4 : 0.6, and the right

lung into upper, middle, and lower lobes with a volume ratio of approximately 0.25 : 0.15 : 0.6, we divide the entire CT volume into three distinct parts based on the ratios of 0.25 : 0.15 : 0.6. Following this, the total budget of slice selection $m$ is distributed across these parts in accordance with the same proportion. CSS is used to select a slice subset in each part. The hyperparameters set for CSS and training the diagnostic model can be found in Appendix B. In the efficiency evaluation experiments, we compared the inference times of different methods while ensuring the same memory usage.

### 4.3 Comparison with Other Compression Methods

In three diagnostic tasks, we compare the performance of diagnostic models when applied with different compression methods, including three interpolation-based methods: Projection Interpolation (PI) [13], Linear Interpolation (LI) [16] and Spline Interpolation (SI) [2], two slice selection-based methods: Subset Slice Selection (SSS) [26] and Even Slice Selection (ESS) [27], and full CT volume method. The implementation of the above methods can be found in Appendix C.

To ensure a fair comparison, we set the number of slices after compression to 64 for each method. The experimental results are shown in Table 1, from which we have several observations: (1) The performance of our method generally surpasses that of other compression methods, and it achieves performance levels very close to those of the full CT volume method across three tasks; (2) On Tasks 1 and 2, the performance of the second-best SI method is comparable to that of our method. On task 3, only our method achieves accuracy and recall above 90%. This may be attributed to the fact that Tasks 1 and 2 involve diagnosing pneumonia, where the lesion areas are larger and more distinct, thus easier to differentiate from normal cases. In contrast, Task 3 focuses on detecting adenocarcinoma, where the lesion areas are smaller and morphologically complex, making accurate detection challenging. The superior performance on difficult tasks demonstrates the superiority of our method.

### 4.4 Exploration Under Extreme Compression

We explore the accuracy and recall of the diagnostic model applied with CSS under extreme compression, that is, selecting 32, 16, 8, and 4 CT slices from each case. We compare CSS not only with the

**Table 1: Comparison with other methods. The best performance is bold, and the second best performance is underlined except for the method of full CT volume. Each result shows mean accuracy and standard deviation over 5-fold cross-validation.**

| Method | Task 1 | | Task 2 | | Task 3 | |
|---|---|---|---|---|---|---|
| | Accuracy | Recall | Accuracy | Recall | Accuracy | Recall |
| Full CT | $99.12_{\pm0.31}$ | $99.15_{\pm0.35}$ | $99.35_{\pm0.41}$ | $99.30_{\pm0.38}$ | $92.47_{\pm0.60}$ | $92.10_{\pm0.65}$ |
| PI [13] | $90.36_{\pm0.34}$ | $89.02_{\pm0.41}$ | $88.92_{\pm0.40}$ | $89.00_{\pm0.30}$ | $84.59_{\pm0.47}$ | $83.50_{\pm0.52}$ |
| LI [16] | $78.52_{\pm2.54}$ | $77.89_{\pm2.67}$ | $73.67_{\pm2.03}$ | $71.70_{\pm2.30}$ | $68.81_{\pm3.80}$ | $67.54_{\pm4.81}$ |
| SI [2] | $98.78_{\pm0.34}$ | $98.20_{\pm0.37}$ | $98.33_{\pm0.42}$ | $98.30_{\pm0.40}$ | $87.24_{\pm0.40}$ | $86.94_{\pm0.37}$ |
| SSS [26] | $96.89_{\pm0.56}$ | $96.28_{\pm0.62}$ | $95.23_{\pm0.97}$ | $95.12_{\pm0.99}$ | $89.34_{\pm0.90}$ | $89.67_{\pm1.24}$ |
| ESS [27] | $96.42_{\pm0.86}$ | $96.85_{\pm0.78}$ | $94.29_{\pm1.37}$ | $94.50_{\pm1.70}$ | $88.56_{\pm1.56}$ | $86.78_{\pm1.65}$ |
| CSS (Ours) | $98.89_{\pm0.17}$ | $99.05_{\pm0.18}$ | $98.65_{\pm0.23}$ | $98.82_{\pm0.29}$ | $91.88_{\pm0.43}$ | $90.85_{\pm0.54}$ |

**Table 2: Comparison under extreme compression. The best performance under different compression levels is bold. Each result shows mean accuracy and standard deviation over 5-fold cross-validation.**

| Number | Method | Task 1 | | Task 2 | | Task 3 | |
|---|---|---|---|---|---|---|---|
| | | Accuracy | Recall | Accuracy | Recall | Accuracy | Recall |
| 32 | SI | $97.99_{\pm0.85}$ | $98.06_{\pm0.54}$ | $96.73_{\pm0.95}$ | $97.10_{\pm0.79}$ | $83.63_{\pm1.24}$ | $83.49_{\pm1.33}$ |
| | CSS | $98.12_{\pm0.55}$ | $98.19_{\pm0.36}$ | $97.89_{\pm0.69}$ | $97.74_{\pm0.77}$ | $90.01_{\pm0.97}$ | $90.27_{\pm0.81}$ |
| 16 | SI | $96.99_{\pm1.55}$ | $97.21_{\pm1.10}$ | $93.87_{\pm2.27}$ | $94.07_{\pm2.03}$ | $76.71_{\pm2.70}$ | $76.90_{\pm2.56}$ |
| | CSS | $97.26_{\pm1.54}$ | $97.58_{\pm1.21}$ | $94.00_{\pm2.15}$ | $94.28_{\pm2.04}$ | $85.60_{\pm2.37}$ | $85.54_{\pm2.40}$ |
| 8 | SI | $95.96_{\pm1.19}$ | $95.88_{\pm1.22}$ | $92.09_{\pm2.77}$ | $92.55_{\pm2.49}$ | $67.34_{\pm3.83}$ | $67.68_{\pm3.41}$ |
| | CSS | $96.57_{\pm1.24}$ | $96.89_{\pm1.27}$ | $93.21_{\pm3.72}$ | $93.02_{\pm3.44}$ | $82.55_{\pm2.32}$ | $82.63_{\pm2.17}$ |
| 4 | SI | $93.86_{\pm2.24}$ | $94.24_{\pm1.89}$ | $90.03_{\pm2.25}$ | $90.78_{\pm2.30}$ | $64.41_{\pm3.70}$ | $64.57_{\pm3.54}$ |
| | CSS | $94.90_{\pm2.12}$ | $95.19_{\pm2.01}$ | $91.17_{\pm2.54}$ | $91.76_{\pm2.51}$ | $78.64_{\pm3.19}$ | $78.89_{\pm2.98}$ |
| Full CT | | $99.12_{\pm0.31}$ | $99.15_{\pm0.35}$ | $99.35_{\pm0.41}$ | $99.30_{\pm0.38}$ | $92.47_{\pm0.60}$ | $92.10_{\pm0.65}$ |

full CT volume method but also with the second-best performing method described in Section 4.3, *i.e.,* SI method [2]. The experimental results are shown in Table 2, from which we have several observations: (1) On Tasks 1 and 2, the performance of models applied with two compression methods is similar, while on Task 3, our method shows a significant improvement over the SI method. This conclusion aligns with the second point discussed in Sec. 4.3; (2) On Task 3, as the number of target slices after compression decreases, the performance improvement of our method becomes increasingly significant compared to the SI method. For instance, at *Number* = 32 and 4, the recall increased by 6.78% and 14.32%, respectively, demonstrating that under extreme compression scenarios, our method holds a greater advantage over other methods.

### 4.5 Comparison Before and After Re-detection

When selecting 8 and 4 CT slices, we verify the effectiveness of the uncertainty strategy by comparing the performance before and after the re-detection. We assume all low-confidence cases will be correctly diagnosed during the re-detection phase. The experimental results are shown in Table 3, from which we observe that the accuracy and recall improve after using the uncertainty strategy to filter out cases for re-detection. In particular, on Task 3, recall improves by 6.17% and 4.14% when retaining 8 and 4 slices,

respectively. This proves that low-confidence cases do contain most of the misdiagnosed cases, which can provide reference value for actual disease screening tasks.

**Table 3: Comparison before and after re-detection using uncertainty strategy.**

| Task | Number | Re-detection | Accuracy | Recall |
|---|---|---|---|---|
| Task 1 | 8 | Before | 96.57 | 96.89 |
| | | After | 97.24(↑ 0.67) | 97.88(↑ 0.99) |
| | 4 | Before | 94.90 | 95.19 |
| | | After | 96.06(↑ 1.16) | 96.41(↑ 1.22) |
| Task 2 | 8 | Before | 93.21 | 93.02 |
| | | After | 94.27(↑ 1.06) | 94.59(↑ 1.57) |
| | 4 | Before | 91.17 | 91.76 |
| | | After | 93.04(↑ 1.87) | 93.18(↑ 1.42) |
| Task 3 | 8 | Before | 82.55 | 82.63 |
| | | After | 88.53(↑ 5.98) | 88.80(↑ 6.17) |
| | 4 | Before | 78.64 | 78.89 |
| | | After | 82.77(↑ 4.13) | 83.03(↑ 4.14) |

## 4.6 Efficiency Evaluation

We compared the inference time of CSS and the full CT method to evaluate the efficiency under different compression rates. The results are shown in Table 4, from which we can see that when retaining only 4 slices per CT volume, the inference time of our method is approximately 4.5% of that of the full CT approach.

**Table 4: Inference time under different compression ratios. The time shown in the table is the inference time of** 50 **cases.**

| Method | Number | Task 1 | Task 2 | Task 3 |
|--------|--------|--------|--------|--------|
| CSS | 32 | $38.74s$ | $38.91s$ | $38.70s$ |
| | 16 | $20.40s$ | $21.27s$ | $20.82s$ |
| | 8 | $11.39s$ | $12.06s$ | $11.58s$ |
| | 4 | $7.69s$ | $8.55s$ | $8.04s$ |
| Full CT | | $169.24s$ | $189.05s$ | $179.25s$ |

## 5 CONCLUSION

In this paper, we propose a novel hierarchical approach based on CT data for accurate and efficient screening of pulmonary diseases. The core insight is leveraging a CTVC method CSS to select representative slices from full CT volumes for coarse detection. During the inference phase, we adopt an uncertainty strategy to identify cases with low diagnostic confidence, which can be referred to radiologists for re-detection in clinical practice. Extensive experimental results show that our approach achieves comparable performance while requiring approximately 4.5% of the time the counterparts need to process full CT volumes. Besides, we also found that our approach outperforms previous SOTA CTVC methods in retaining crucial information after compression.

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

## A DETAILS OF CLASSIFICATION MODEL

ResNet-34 is used to extract image features. Specifically, we replace the $1000 - d$ Fully-Connected (FC) layer in the penultimate layer with a $512 - d$ FC layer and delete the last SoftMax layer. The output of ResNet-34 is used as input to the Transformer Encoder, where both the layer number and the head number of the Transformer Encoder are set to 6. Finally, we perform an FC layer and Softmax on the output of the transformer-based network to predict the probability of disease. The prediction loss function is a cross-entropy loss function.

## B HYPERPARAMETER SETTINGS

For CSS, we set the hyperparameters $k = 10$, $\alpha = 0.5$, $m_{\text{reg}} = 0.9$, $\lambda = 0.5$ and $t = 10$. During the training stage of the diagnostic model, we use adaptive moment estimation (Adam) with an initial learning rate $1e^{-3}$ to optimize the network. We set different batch sizes for experiments with different numbers of compressed selected

slices. The more slices selected for each case, the more GPU memory each case occupies; thus, the batch size should be smaller. In detail, $batchsize = 1, 4, 8, 16, 16$ is set for $m = 64, 32, 16, 8, 4$ respectively. For a fair comparison, the training epoch of all experiments is set to 400.

## C   IMPLEMENTATION OF OTHER COMPRESSION METHODS

Interpolation-based methods calculate the new pixel value in the compressed image based on the values of surrounding pixels on the Z-axis. As for the slice selection-based method, SSS selects equal slices from the first, middle and last position of the CT volume, whereas ESS selects one after every specific number of slices. As a baseline method, the full CT volume method utilizes the whole CT dataset for training the classification model without compression.