# OpenReview forum: "Comprehensive Subset Selection for CT Volume Compression to Improve Pulmonary Disease Screening Efficiency"
_KDD.org/2024/Workshop/AIDSH — KDD-AIDSH 2024 Poster_

### Official Review · Reviewer_b2Un · 2024-06-11
**The authors propose an efficient pulmonary disease screening approach via analyzing a subset of CT scans selected using K means clustering. They also validated the effectiveness and efficiency of the proposed method using three binary classification tasks on two publicly available datasets. The method for pulmonary disease screening is somewhat novel. The paper is well written and easy to understood. However, the binary classification tasks (Common Pneumonia vs Normal, Novel Coronavirus Pneumonia vs. Normal, and Adenocarcinoma vs. Normal) are too easy and far away from clinical practice.**

**Rating:** 6
**Confidence:** 4

**Review:**

The method for pulmonary disease screening is somewhat novel. The paper is well written and easy to understood.

Major concerns:

1. The binary classification tasks (Common Pneumonia vs Normal, Novel Coronavirus Pneumonia vs. Normal, and Adenocarcinoma vs. Normal) are too easy and far away from clinical practice. I believe the classification tasks are designed to tailor the proposed method. Whether the proposed method work for classifying novel coronavirus pneumonia and common pneumonia or not? Please clarify.

2. The authors conclude that “the screening method can achieve similar performance while requiring approximately 4.5% of the time needed by the method using full CT volumes”. It seems this conclusion is overclaimed. The model based on four selected CT slices need the 4.5% of time of the method using full CT volumes is need and only achieves an accuracy of 78.64% for task 3.

3. (Experimental Settings): It is confusing how to divide the entire CT volume into three distinct parts and what the three parts are.

4. It is confusing whether the inference time includes the Representation Learning (feature extraction using the pretrained MedCLIP-ViT). The time of feature extract should be included when application of the proposed method.
5. Which model is used for analyzing full CT scans?

---

### Official Review · Reviewer_UkmQ · 2024-06-18
**Reviews for Hierarchically Efficient Pulmonary Disease Screening via CT Compression**

**Rating:** 7
**Confidence:** 4

**Review:**

Due to the large pixel size of CT data and the time-consuming and resource-intensive screening process for pulmonary diseases, this work proposes the CTVC method to achieve accurate and efficient results. In terms of accuracy, it outperforms previous state-of-the-art CTVC methods. Regarding time consumption, it requires only approximately 4.5% of the time needed by other methods to process full CT volumes.

This work employs MedCLIP-ViT to learn representations of CT volumes, performs K-means clustering to select representative slices, and adds regularizations to the objective function to ensure the selected slices are as diverse as possible. Subsequently, a ResNet-34 model is trained for disease classification tasks.
Each part of the article is written clearly, and the sentences are concise.
However, this work also has some shortcomings:
1. Lack of innovation: The main advantage of the article's information compression, compared to other methods, is that it reduces the likelihood of omitting slices containing crucial indicators. This goal is primarily achieved with the help of the MedCLIP-ViT model. Additionally, the selection of slices is based on common methods such as K-means clustering results, KNN methods, and regularization constraints.
2. Unclear "hierarchical approach": The "hierarchical approach" mentioned in the article does not clearly specify the levels involved.

---

### Decision · Program_Chairs · 2024-06-28

Accept (Poster)